# MAKING SUBSTITUTE MODELS MORE BAYESIAN CAN ENHANCE TRANSFERABILITY OF ADVERSARIAL EXAMPLES

**Qizhang Li**[1,2]**, Yiwen Guo**[3*]**, Wangmeng Zuo**[1]**, Hao Chen**[4]
[1]Harbin Institute of Technology, [2]Tencent Security Big Data Lab, [3]Independent Researcher, [4]UC Davis
`{liqizhang95,guoyiwen89}@gmail.com`  `wmzuo@hit.edu.cn`  `chen@ucdavis.edu`

## ABSTRACT

The transferability of adversarial examples across deep neural networks (DNNs) is the crux of many black-box attacks. Many prior efforts have been devoted to improving the transferability via increasing the diversity in inputs of some substitute models. In this paper, by contrast, we opt for the diversity in substitute models and advocate to attack a Bayesian model for achieving desirable transferability. Deriving from the Bayesian formulation, we develop a principled strategy for possible finetuning, which can be combined with many off-the-shelf Gaussian posterior approximations over DNN parameters. Extensive experiments have been conducted to verify the effectiveness of our method, on common benchmark datasets, and the results demonstrate that our method outperforms recent state-of-the-arts by large margins (roughly $19\%$ absolute increase in average attack success rate on ImageNet), and, by combining with these recent methods, further performance gain can be obtained. Our code: https://github.com/qizhangli/MoreBayesian-attack.

## 1 INTRODUCTION

The adversarial vulnerability of deep neural networks (DNNs) has attracted great attention (Szegedy et al., 2014; Goodfellow et al., 2015; Papernot et al., 2016; Carlini & Wagner, 2017; Madry et al., 2018; Athalye et al., 2018). It has been demonstrated that the prediction of state-of-the-art DNNs can be arbitrarily altered by adding perturbations, even imperceptible to human eyes, to their inputs.

Threat models concerning adversarial examples can be divided into white-box and black-box ones according to the amount of information (of victim models) being exposed to the attacker. In black-box attacks, where the attacker can hardly get access to the architecture and parameters of the victim model, the transferability of adversarial examples is often relied on, given the fact that adversarial examples crafted on a substitute model can sometimes fool other models as well. However, such methods also suffer from considerable failure rate when the perturbation budget is small. Thus, much recent effort has been devoted to improving the black-box transferability of adversarial examples, and a variety of transfer-based attacks have been proposed.

Assuming that the substitute model was pre-trained and given, most of the recent research focused only on improving the backpropagation process when issuing attacks, yet little attention has been paid to possible training or finetuning of the substitute model. In this paper, we shall focus more on the training process, and for which we advocate to perform in a Bayesian manner, in order to issue more powerful transfer-based attacks. By introducing probability measures to weights and biases of the substitute model, all these parameters are represented under assumptions of some distributions to be learned. In this way, an ensemble of infinitely many DNNs (that are jointly trained in our view) can be obtained from a single run of training. Adversarial examples are then crafted by maximizing average prediction loss over such a distribution of models, which is a referred to as posterior learned in the Bayesian manner. Experiments on attacking a variety of CIFAR-10 (Krizhevsky & Hinton, 2009) and ImageNet (Russakovsky et al., 2015) victim models have been performed, and we show that the proposed method outperforms state-of-the-arts considerably. Moreover, our method can be conjugated with existing methods easily and reliably for further improving the attack performance.

---

[*]Work was done under the supervision of Yiwen Guo who is the corresponding author.

## 2 BACKGROUND AND RELATED WORK

### 2.1 ADVERSARIAL ATTACKS

**White-box attacks.** Given full knowledge of the architecture and parameters of a victim model, white-box attacks are typically performed via utilizing some loss gradient with respect to the model inputs. For instance, given a normal sample $(\mathbf{x}, y)$ and a model $f_\mathbf{w} : \mathbb{R}^n \to \mathbb{R}^c$ that is trained to classify $\mathbf{x}$ into $y \in R^c$, it is a popular choice to craft the adversarial example $\mathbf{x} + \mathbf{\Delta x}$ within an $\ell_p$ bounded small region of $\mathbf{x}$, by maximizing the prediction loss, *i.e.*, $\max_{\|\mathbf{\Delta x}\|_p \leq \epsilon} L(\mathbf{x} + \mathbf{\Delta x}, y, \mathbf{w})$, where $\epsilon$ is the perturbation budget. FGSM proposed to calculate $\epsilon \cdot \mathrm{sgn}(\nabla_\mathbf{x} L(\mathbf{x}, y, \mathbf{w}))$ for $\mathbf{\Delta x}$ in the $p = \infty$ setting (Goodfellow et al., 2015), and the iterative variants of FGSM, *e.g.*, I-FGSM (Kurakin et al., 2017) and PGD (Madry et al., 2018) can be more powerful.

**Black-box attacks.** Black-box attacks are more challenging compared to the white-box attacks. Many existing methods largely rely on the transferability of adversarial examples, *i.e.*, adversarial examples crafted on one classification model can generally succeed in attacking some other victim models as well. It is normally assumed to be able to query the victim model to annotate training samples, or be possible to collect a pre-trained source model that is trained to accomplish the same task as the victim models. Aiming at enhancing the adversarial transferability, methods have been proposed to modify the backpropagation computation, see for example the skip gradient method (SGM) (Wu et al., 2020), the linear backpropagation (LinBP) method (Guo et al., 2020), the intermediate-level attack (ILA) (Huang et al., 2019), and ILA++ (Li et al., 2020a; Guo et al., 2022). It is also widely adopted to increase the diversity in inputs (Xie et al., 2019; Dong et al., 2019; Lin et al., 2019; Wang et al., 2021). In this paper, we consider the diversity from another perspective, the substitute model(s), and we introduce a Bayesian approximation for achieving this.

**Ensemble-based attacks.** Our method can be equivalent to utilizing an ensemble of infinitely many substitute models with different parameters for performing attacks. There exists prior work that also took advantage of multiple substitute models. For instance, Liu et al. (2017) proposed to generate adversarial examples on an ensemble of multiple models that differ in their architectures. Li et al. (2020b) proposed ghost network for gaining transferability, using dropout and skip connection erosion to obtain multiple models. Following the spirit of stochastic variance reduced gradient (Johnson & Zhang, 2013), Xiong et al. (2022) proposed stochastic variance reduced ensemble (SVRE) to reduce the variance of gradients of different substitute models. From a geometric perspective, Gubri et al. (2022b) suggested finetuning with a constant and high learning rate for collecting multiple models along the training trajectory, on which the ensemble attack was performed. Another method collected substitute models by using cSGLD (Gubri et al., 2022a), which is more related to our work, but being different in the sense of posterior approximation and sampling strategy. We will provide detailed comparison in Section 4.2.

### 2.2 BAYESIAN DNNs

If a DNN is viewed as a probabilistic model, then the training of its parameters $\mathbf{w}$ can be regarded as maximum likelihood estimation or maximum a posterior estimation (with regularization). Bayesian deep learning opts for estimating a posterior of the parameter given data at the same time. Prediction of any new input instance is given by taking expectation over such a posterior. Since DNNs normally involves a huge number of parameters, making the optimization of Bayesian model more challenging than in shallow models, a series of studies have been conducted and many scalabble approximations have been developed. Effective methods utilize variational inference (Graves, 2011; Blundell et al., 2015; Kingma et al., 2015; Khan et al., 2018; Zhang et al., 2018; Wu et al., 2018; Osawa et al., 2019; Dusenberry et al., 2020) dropout inference (Gal & Ghahramani, 2016; Kendall & Gal, 2017; Gal et al., 2017), Laplace approximation (Kirkpatrick et al., 2017; Ritter et al., 2018; Li, 2000), or SGD-based approximation (Mandt et al., 2017; Maddox et al., 2019; 2021; Wilson & Izmailov, 2020). Taking SWAG (Maddox et al., 2019) as an example, which is an SGD-based approximation, it approximates the posterior using a Gaussian distribution with the stochastic weight averaging (SWA) solution as its first raw moment and the composition of a low rank matrix and a diagonal matrix as its second central moment. Our method is developed in a Bayesian spirit and we shall discuss SWAG thoroughly in this paper. Due to the space limit of this paper, we omit detailed introduction of these methods and encourage readers to check references if needed.

The robustness of Bayesian DNNs has also been studied over the last few years. In addition to the probabilistic robustness/safety measures of such models (Cardelli et al., 2019; Wicker et al., 2020), attacks have been adapted (Liu et al., 2018b; Yuan et al., 2020) to testing the robustness in Bayesian settings. Theoretical studies have also been made (Gal & Smith, 2018). Although Bayesian models are suggested to be more robust (Carbone et al., 2020; Li et al., 2019), adversarial training has also been proposed for them, as in Liu et al. (2018b)'s work. Yet, to the best of our knowledge, these studies did not pay attention to adversarial transferability.

## 3 TRANSFER-BASED ATTACK AND BAYESIAN SUBSTITUTE MODELS

An intuition for improving the transferability of adversarial examples suggests improving the diversity during backpropagation. Prior work has tried increasing input diversity (Xie et al., 2019; Dong et al., 2019; Lin et al., 2019; Wang et al., 2021) and has indeed achieved remarkable improvements. In this paper we consider model diversity. A straightforward idea seems to train a set of models with diverse architecture or from different initialization. If all models (including the victim models) that are trained to accomplish the same classification task follow a common distribution, then training multiple substitute models seems to perform multiple point estimates with maximum likelihood estimation. The power of performing attacks on the ensemble of these models may increase along with the number of substitute models. However, the time complexity of such a straightforward method is high, and it scales with the number of substitute models that could be trained. Here we opt for an Bayesian approach to address this issue and the method resembles performing transfer-based attack on an ensemble of infinitely many DNNs.

### 3.1 GENERATE ADVERSARIAL EXAMPLES ON A BAYESIAN MODEL

Bayesian learning aims to discover a distribution of likely models instead of a single one. Given a posterior distribution over parameters $p(\mathbf{w}|\mathcal{D}) \propto p(\mathcal{D}|\mathbf{w})p(\mathbf{w})$, where $\mathcal{D}$ is the dataset, we can predict the label of a new input $\mathbf{x}$ by Bayesian model averaging, *i.e.*,

$$p(y|\mathbf{x}, \mathcal{D}) = \int_{\mathbf{w}} p(y|\mathbf{x}, \mathbf{w})p(\mathbf{w}|\mathcal{D})d\mathbf{w}, \tag{1}$$

where $p(y|\mathbf{x}, \mathbf{w})$ is the likelihood, sometimes also referred to as the predictive distribution (Izmailov et al., 2021; Lakshminarayanan et al., 2017) for a given $\mathbf{w}$, which is obtained from the DNN output with the assistance of the softmax function. To perform attack on such a Bayesian model, a straightforward idea is to solve the following optimization problem (Liu et al., 2018b; Carbone et al., 2020):

$$\underset{\|\mathbf{\Delta x}\|_p \leq \epsilon}{\arg\min} \, p(y|\mathbf{x} + \mathbf{\Delta x}, \mathcal{D}) = \underset{\|\mathbf{\Delta x}\|_p \leq \epsilon}{\arg\min} \int_{\mathbf{w}} p(y|\mathbf{x} + \mathbf{\Delta x}, \mathbf{w})p(\mathbf{w}|\mathcal{D})d\mathbf{w}. \tag{2}$$

Obviously, it is intractable to perform exact inference on DNNs using Eq. (2), since there are a very large number of parameters. A series of methods aim to address this, and, as in prior work, we adopt the Monte Carlo sampling method to approximate the integral, *i.e.*, $p(y|\mathbf{x}, \mathcal{D}) \approx \frac{1}{M} \sum_i p(y|\mathbf{x}, \mathbf{w}_i)$, where a set of $M$ models each of which parameterized by $\mathbf{w}_i$ are sampled from the posterior $p(\mathbf{w}|\mathcal{D})$. One can then solve Eq. (2) by performing attacks on the ensemble of these models,

$$\underset{\|\mathbf{\Delta x}\|_p \leq \epsilon}{\arg\min} \frac{1}{M} \sum_{i=1}^{M} p(y|\mathbf{x} + \mathbf{\Delta x}, \mathbf{w}_i) = \underset{\|\mathbf{\Delta x}\|_p \leq \epsilon}{\arg\max} \frac{1}{M} \sum_{i=1}^{M} L(\mathbf{x} + \mathbf{\Delta x}, y, \mathbf{w}_i), \text{s.t.} \, \mathbf{w}_i \sim p(\mathbf{w}|\mathcal{D}). \tag{3}$$

where $L(\cdot, \cdot, \mathbf{w}_i)$ is a function evaluating prediction loss of a DNN model parameterized by $\mathbf{w}_i$. With iterative optimization methods, *e.g.*, I-FGSM and PGD, different sets of models can be sampled at different iterations, as if there exist infinitely many substitute models.

### 3.2 THE BAYESIAN FORMULATION AND POSSIBLE FINETUNING

In this subsection, we discuss the way of obtaining the Bayesian posterior. Following prior work, we consider a threat model in which finetuning a source model is sometimes possible on benchmark datasets collected for the same task as that of the victim models, though it is feasible to approximate the posterior without taking special care of the training process (Gal & Ghahramani, 2016).

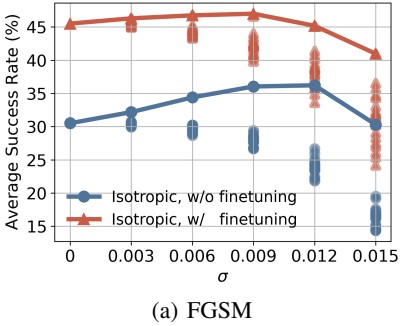 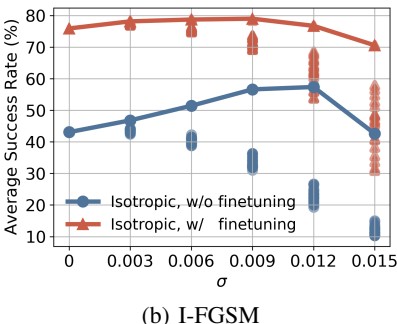

(a) FGSM  (b) I-FGSM

Figure 1: How the (a) FGSM transferability, and (b) I-FGSM transferability change with different choices of $\sigma$ for the isotropic Gaussian posterior on CIFAR-10. The dots in the plots represent the transferability of adversarial examples crafted by a single sample in the posterior. We performed $\ell_\infty$ attacks under $\epsilon = 4/255$. Best viewed in color.

To get started, we simply assume that the posterior is an isotropic Gaussian $\mathcal{N}(\hat{\mathbf{w}}, \sigma^2 \mathbf{I})$, where $\hat{\mathbf{w}}$ is the parameter to be trained and $\sigma$ is a positive constant for controlling the diversity of distribution. The rationality of such an assumption of isotropic posterior comes from the fact that the distribution of victim models is unknown and nearly infeasible to estimate in practice, and higher probability of the victim parameters in the posterior may imply stronger transferability. We thus encourage exploration towards all directions (departed from $\hat{\mathbf{w}}$) of equal importance in the first place. Discussions with a more practical assumption of the posterior will be given in the next section.

Optimization of the Bayesian model can be formulated as

$$\max_{\hat{\mathbf{w}}} \frac{1}{N} \sum_{i=1}^{N} \mathbb{E}_{\mathbf{w} \sim \mathcal{N}(\hat{\mathbf{w}}, \sigma^2 \mathbf{I}))} p(y_i | \mathbf{x}_i, \mathbf{w}). \tag{4}$$

We can further reformulate Eq. (4) into

$$\min_{\hat{\mathbf{w}}} \frac{1}{MN} \sum_{i=1}^{N} \sum_{j=1}^{M} L(\mathbf{x}_i, y_i, \hat{\mathbf{w}} + \Delta \mathbf{w}_j), \text{ s.t. } \Delta \mathbf{w}_j \sim \mathcal{N}(\mathbf{0}, \sigma^2 \mathbf{I}). \tag{5}$$

by adopting Monte Carlo sampling. The computational complexity of the objective in Eq. (5) seems still high, thus we focus on the worst-case parameters from the posterior, whose loss in fact bounds the objective from below. The optimization problem then becomes:

$$\min_{\hat{\mathbf{w}}} \max_{\Delta \mathbf{w}} \frac{1}{N} \sum_{i=1}^{N} L(\mathbf{x}_i, y_i, \hat{\mathbf{w}} + \Delta \mathbf{w}), \text{ s.t. } \Delta \mathbf{w} \sim \mathcal{N}(\mathbf{0}, \sigma^2 \mathbf{I}) \text{ and } p(\Delta \mathbf{w}) \geq \varepsilon, \tag{6}$$

where $\varepsilon$ controls the confidence region of the Gaussian posterior. With Taylor's theorem, we further approximate Eq. (6) with

$$\min_{\hat{\mathbf{w}}} \max_{\Delta \mathbf{w}} \frac{1}{N} \sum_{i=1}^{N} L(\mathbf{x}_i, y_i, \hat{\mathbf{w}}) + \nabla_{\hat{\mathbf{w}}} L(\mathbf{x}_i, y_i, \hat{\mathbf{w}})^T \Delta \mathbf{w}, \text{ s.t. } \Delta \mathbf{w} \sim \mathcal{N}(\mathbf{0}, \sigma^2 \mathbf{I}) \text{ and } p(\Delta \mathbf{w}) \geq \varepsilon. \tag{7}$$

As $\Delta \mathbf{w}$ is sampled from a zero-mean isotropic Gaussian distribution, the inner maximization problem has an analytic solution, which is $\Delta \mathbf{w}^* = \frac{1}{N} \sum_{i=1}^{N} \lambda_{\varepsilon,\sigma} \nabla_{\hat{\mathbf{w}}} L(\mathbf{x}_i, y_i, \hat{\mathbf{w}}) / \|\nabla_{\hat{\mathbf{w}}} L(\mathbf{x}_i, y_i, \hat{\mathbf{w}})\|$. $\lambda_{\varepsilon,\sigma}$ is computed with the probability density of the zero-mean isotropic Gaussian distribution. Thereafter, the outer gradient for solving Eq. (7) is $\frac{1}{N} \sum_{i=1}^{N} \nabla_{\hat{\mathbf{w}}} L(\mathbf{x}_i, y_i, \hat{\mathbf{w}}) + \mathbf{H} \Delta \mathbf{w}^*$, which involves second-order partial derivatives in the Hessian matrix $\mathbf{H}$ which can be approximately calculated using the finite difference method. More specifically, the gradient is estimated using $\frac{1}{N} \sum_{i=1}^{N} \nabla_{\hat{\mathbf{w}}} L(\mathbf{x}_i, y_i, \hat{\mathbf{w}}) + (1/\gamma)(\nabla_{\hat{\mathbf{w}}} L(\mathbf{x}_i, y_i, \hat{\mathbf{w}} + \gamma \Delta \mathbf{w}^*) - \nabla_{\hat{\mathbf{w}}} L(\mathbf{x}_i, y_i, \hat{\mathbf{w}}))$, where $\gamma$ is a small positive constant.

A quick experiment on CIFAR-10 (Krizhevsky & Hinton, 2009) using ResNet-18 (He et al., 2016) was performed to evaluate the effectiveness of such finetuning. Introduction to victim models and detailed experimental settings are deferred to Section 4.1. In Figure 1, we compare attack transferability using the conventional deterministic formulation and the Bayesian formulation, by applying FGSM and I-FGSM for single-step and multi-step attacks, respectively. We evaluated with different choices of the posterior covariance by varying the value of $\sigma$. Notice that, as has been mentioned,

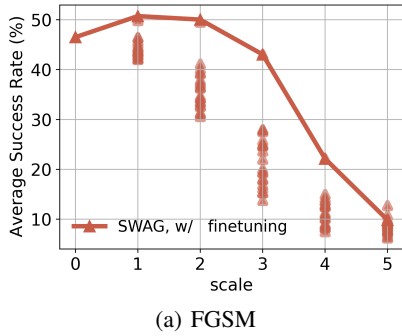 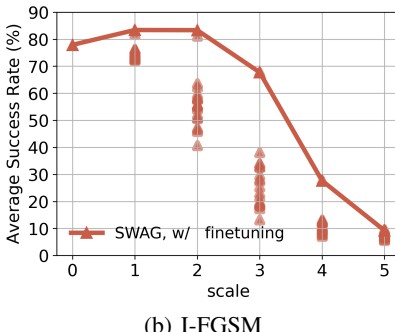

(a) FGSM             (b) I-FGSM

Figure 2: How the (a) FGSM transferability, and (b) I-FGSM transferability change with different scales for the covariance of SWAG posterior on CIFAR-10. The dots in the plots represent the transferability of adversarial examples crafted by a single model in the posterior. We perform $\ell_\infty$ attacks under $\epsilon = 4/255$. Best viewed in color.

the posterior can also be given even without any finetuning, and we achieve this via directly applying the pre-trained parameters to $\hat{\mathbf{w}}$ in $\mathcal{N}(\hat{\mathbf{w}}, \sigma^2 \mathbf{I})$. Attacking such a "Bayesian" model can be equivalent to attacking the deterministic model when $\sigma \to 0$. With $\sigma$ being strictly greater than 0, one can perform attack in a way as described in Section 3.1. Apparently, more significant adversarial transferability can be achieved by introducing the Bayesian formulation (along with $\sigma > 0$), while the substitute model itself is suggested to be more robust with such a Bayesian formulation (Li et al., 2019; Carbone et al., 2020). In Figure 1, see the blue curves that are obtained without any finetuning, the best performance is achieved with $\sigma = 0.012$, showing $+5.71\%$ and $+14.30\%$ absolute gain in attack success rate caused by attacks with FGSM and I-FGSM, respectively.

It can further be observed that obvious improvement (over $+21\%$ absolute increase in the I-FGSM success rate) can be achieved by introducing the additional finetuning, though attacking the pre-trained parameters in a Bayesian manner already outperforms the baseline considerably. Detailed performance on each victim model is reported in Table 1. Conclusions on ImageNet is consistent with that on CIFAR-10, see Appendix A.

The (possible) finetuning improves the performance of unsatisfactory models that could be sampled. For instance, with $\sigma = 0.009$, the worst performance in 100 model samples from the posterior shows an test-set prediction accuracy of $90.59\%$ while the model at population mean shows an accuracy of $91.68\%$. After finetuning, both results increase, to $91.91\%$ and $92.65\%$, respectively, which means finetuning improves the test-set performance of Bayesian samples and this could be beneficial to adversarial transferability.

Table 1: Comparing transferability of FGSM and I-FGSM adversarial examples generated on a deterministic substitute model and the Bayesian substitute model (with or without additional finetuning) under the $\ell_\infty$ constraint with $\epsilon = 4/255$ on CIFAR-10. The architecture of the substitute models is ResNet-18, and "average" was calculated over all six victim models except for ResNet-18. We performed 10 runs of the experiment and report the average performance in the table.

|  |  | fine-tune | ResNet-18 | VGG-19 | WRN | ResNeXt | DenseNet | PyramidNet | GDAS | Average |
|---|---|---|---|---|---|---|---|---|---|---|
| FGSM | - | ✗ | 84.76% | 33.95% | 36.10% | 36.89% | 34.30% | 13.21% | 28.60% | 30.51% |
|  | Isotropic | ✗ | 85.14% | 40.41% | 43.33% | 43.97% | 41.17% | 14.92% | 33.51% | 36.22% |
|  |  | ✓ | 85.92% | 54.06% | 55.50% | 57.40% | 53.13% | 19.23% | 42.69% | 47.00% |
|  | +SWAG | ✓ | 85.99% | 59.01% | 59.64% | 61.35% | 57.67% | 21.22% | 45.57% | 50.74% |
| I-FGSM | - | ✗ | 100.00% | 37.51% | 57.01% | 57.95% | 52.84% | 12.70% | 40.32% | 43.06% |
|  | Isotropic | ✗ | 100.00% | 52.30% | 73.46% | 75.38% | 69.55% | 19.61% | 53.85% | 57.36% |
|  |  | ✓ | 100.00% | 78.80% | 91.63% | 92.98% | 90.22% | 41.50% | 78.76% | 78.98% |
|  | +SWAG | ✓ | 100.00% | 85.10% | 94.66% | 94.95% | 92.75% | 48.93% | 83.89% | 83.38% |

## 3.3 FORMULATION WITH IMPROVED APPROXIMATION

In Section 3.2, we have demonstrated the power of adopting Bayesian substitute models for generating transferable adversarial examples, with a relatively strong assumption of the posterior with a presumed isotropic covariance matrix though. Taking one step further, we try to learn the covariance matrix from data in this subsection.

Table 2: Success rates of transfer-based attacks on ImageNet using ResNet-50 as substitute architecture and I-FGSM as the back-end attack, under the $\ell_\infty$ constraint with $\epsilon = 8/255$ in the untargeted setting. "Average" was calculated over all ten victim models except for ResNet-50. We performed 10 runs and report the average performance for each result in the table.

| Method | ResNet-50 | VGG-19 | ResNet-152 | Inception v3 | DenseNet | MobileNet |
|---|---|---|---|---|---|---|
| I-FGSM | **100.00%** | 39.22% | 29.18% | 15.60% | 35.58% | 37.90% |
| TIM (2019) | **100.00%** | 44.98% | 35.14% | 22.21% | 46.19% | 42.67% |
| SIM (2020) | **100.00%** | 53.30% | 46.80% | 27.04% | 54.16% | 52.54% |
| LinBP (2020) | **100.00%** | 72.00% | 58.62% | 29.98% | 63.70% | 64.08% |
| Admix (2021) | **100.00%** | 57.95% | 45.82% | 23.59% | 52.00% | 55.36% |
| TAIG (2022) | **100.00%** | 54.32% | 45.32% | 28.52% | 53.34% | 55.18% |
| ILA++ (2022) | 99.96% | 74.94% | 69.64% | 41.56% | 71.28% | 71.84% |
| LGV (2022) | **100.00%** | 89.02% | 80.38% | 45.76% | 88.20% | 87.18% |
| Ours | **100.00%** | **97.79%** | **97.13%** | **73.12%** | **98.02%** | **97.49%** |

| Method | SENet | ResNeXt | WRN | PNASNet | MNASNet | Average |
|---|---|---|---|---|---|---|
| I-FGSM | 17.66% | 26.18% | 27.18% | 12.80% | 35.58% | 27.69% |
| TIM (2019) | 22.47% | 32.11% | 33.26% | 21.09% | 39.85% | 34.00% |
| SIM (2020) | 27.04% | 41.28% | 42.66% | 21.74% | 50.36% | 41.69% |
| LinBP (2020) | 41.02% | 51.02% | 54.16% | 29.72% | 62.18% | 52.65% |
| Admix (2021) | 30.28% | 41.94% | 42.78% | 21.91% | 52.32% | 42.40% |
| TAIG (2022) | 24.82% | 38.36% | 42.16% | 17.20% | 54.90% | 41.41% |
| ILA++ (2022) | 53.12% | 65.92% | 65.64% | 44.56% | 70.40% | 62.89% |
| LGV (2022) | 54.82% | 71.22% | 75.14% | 46.50% | 84.58% | 72.28% |
| Ours | **85.41%** | **94.16%** | **95.39%** | **77.60%** | **97.15%** | **91.33%** |

There exist dozens of methods for achieving the goal, here we choose SWAG (Maddox et al., 2019) which is a simple and scalable one. It introduces an improved approximation to the posterior over parameters. Gaussian approximation is still considered, and more specifically, the SWA solution (Izmailov et al., 2018) is adopted as its mean and the covariance is decomposed into a low rank matrix and a diagonal matrix, *i.e.*, $\mathbf{w} \sim \mathcal{N}(\mathbf{w}_{\mathrm{SWA}}, \mathbf{\Sigma}_{\mathrm{SWAG}})$, where $\mathbf{\Sigma}_{\mathrm{SWAG}} = \frac{1}{2}(\mathbf{\Sigma}_{\mathrm{diag}} + \mathbf{\Sigma}_{\mathrm{low-rank}})$.

In SWAG, both the mean and the covariance are calculated using temporary models during training, and thus the posterior is estimated from the training dynamics. Recall that the posterior concerned in the previous section is constructed using only a single model sampling, thus it can be readily combined with SWAG for improving the diversity and flexibility. Specifically, since $\mathbf{w}_{\mathrm{SWA}}$ is unknown before training terminates, we optimize models using the same learning objective as in Section 3.2. On this point, the dispersion of $\mathbf{w}$ in the final Bayesian model comes from two independent Gaussian distribution and the covariance matrices are added together, *i.e.*, $\mathbf{w} \sim \mathcal{N}(\mathbf{w}_{\mathrm{SWA}}, \mathbf{\Sigma}_{\mathrm{SWAG}} + \beta\mathbf{I})$, where $\beta = \sigma^2$.

Figure 2 illustrates the attack performance when SWAG is further incorporated for approximating the posterior. It can be seen that our method works favourably well, and further improvements can be achieved comparing to the results in Figure 1. Detailed success rates on each victim model are given in Table 1. Conclusions on ImageNet are the same (see Appendix A). Such empirical improvements indicate that the assumption of a more general Gaussian posterior may still align with the distribution of victim parameters in practice, and SWAG help estimate the posterior appropriately. Thus, without further clarification, we will incorporate SWAG into our method in the following experiments considering the superior performance. Note that since SWAG requires continuous model checkpointing which is normally unavailable without finetuning, we only report the performance of our method with finetuning in the tables. If the collected source model was trained with frequent checkpointing and all these checkpoints are available, this method can be applied without finetuning.

# 4 EXPERIMENTS

We evaluate the effectiveness of our method by comparing it to recent state-of-the-arts in this section.

## 4.1 EXPERIMENTAL SETTINGS

We tested untargeted $\ell_\infty$ attacks in the black-box setting, just like prior work (Dong & Yang, 2019; Lin et al., 2019; Wang et al., 2021; Guo et al., 2020; 2022). A bunch of methods were considered for comparison on CIFAR-10 (Krizhevsky & Hinton, 2009), and ImageNet (Russakovsky et al., 2015), using I-FGSM as the back-end method. On CIFAR-10, we set the perturbation budget to $\epsilon = 4/255$ and used ResNet-18 (He et al., 2016) as source model. While on ImageNet, the perturbation bud-

Table 3: Success rates caused by transfer-based attacks on CIFAR-10 using ResNet-18 as substitute architecture and I-FGSM as the back-end attack, under the $\ell_\infty$ constraint with $\epsilon = 4/255$ in the untargeted setting. "Average" was calculated over all six victim models except for ResNet-18. We performed 10 runs and report the average performance for each result in the table.

| Method | ResNet-18 | VGG-19 | WRN | ResNeXt | DenseNet | PyramidNet | GDAS | Average |
|---|---|---|---|---|---|---|---|---|
| I-FGSM | **100.00%** | 37.51% | 57.01% | 57.95% | 52.84% | 12.70% | 40.32% | 43.06% |
| TIM (2019) | **100.00%** | 39.65% | 58.41% | 59.74% | 54.07% | 13.33% | 40.59% | 44.30% |
| SIM (2020) | **100.00%** | 47.62% | 64.62% | 68.41% | 62.43% | 17.09% | 44.46% | 50.77% |
| LinBP (2020) | **100.00%** | 58.43% | 78.49% | 81.10% | 76.50% | 27.20% | 60.91% | 63.77% |
| Admix (2021) | **100.00%** | 49.17% | 69.94% | 69.95% | 64.65% | 16.90% | 49.50% | 53.35% |
| TAIG (2022) | **100.00%** | 47.20% | 59.70% | 63.18% | 56.83% | 15.29% | 43.92% | 47.69% |
| ILA++ (2022) | **100.00%** | 59.46% | 78.03% | 78.49% | 74.91% | 25.60% | 59.11% | 62.60% |
| LGV (2022) | **100.00%** | 80.62% | 92.52% | 92.67% | 90.44% | 41.50% | 77.28% | 79.17% |
| Ours | **100.00%** | **85.10%** | **94.66%** | **94.95%** | **92.75%** | **48.93%** | **83.89%** | **83.38%** |

get was set to $\epsilon = 8/255$ and used ResNet-50 (He et al., 2016) as the source model. For victim models on CIFAR-10, we chose a ResNet-18 (He et al., 2016), a VGG-19 with batch normalization (Simonyan & Zisserman, 2015), a PyramidNet (Han et al., 2017), GDAS (Dong & Yang, 2019), a WRN-28-10 (Zagoruyko & Komodakis, 2016), a ResNeXt-29 (Xie et al., 2017), and a DenseNet-BC (Huang et al., 2017) [1], following Guo et al. (2020; 2022)'s work. On ImageNet, ResNet-50 (He et al., 2016), VGG-19 (Simonyan & Zisserman, 2015), ResNet-152 (He et al., 2016), Inception v3 (Szegedy et al., 2016), DenseNet-121 (Huang et al., 2017), MobileNet v2 (Sandler et al., 2018), SENet-154 (Hu et al., 2018), ResNeXt-101 (Xie et al., 2017), WRN-101 (Zagoruyko & Komodakis, 2016), PNASNet (Liu et al., 2018a), and MNASNet (Tan et al., 2019) were adopted as victim models [2]. Since these victim models may require different sizes of inputs, we strictly followed their official pre-processing pipeline to obtain inputs of specific sizes. For CIFAR-10 tests, we performed attacks on all test data. For ImageNet, we randomly sampled 5000 test images from a set of the validation data that could be classified correctly by these victim models, and we learned perturbations to these images, following prior work (Huang & Kong, 2022; Guo et al., 2020; 2022). For comparison, we evaluated attack success rates of adversarial examples crafted utilizing different methods on all victim models. Inputs to all models were re-scaled to $[0.0, 1.0]$. Temporary results after each attack iteration were all clipped to this range to guarantee the inputs to DNNs were close to valid images. When establishing a multi-step baseline using I-FGSM we run it for 20 iterations on CIFAR-10 data and 50 iterations on ImageNet data with a step size of $1/255$.

In possible finetuning, we set $\gamma = 0.1/\|\Delta \mathbf{w}^*\|_2$ and a finetuning learning rate of $0.05$ if SWAG was incorporated. We set a smaller finetuning learning rate of $0.001$ if it was not. We use an SGD optimizer with a momentum of $0.9$ and a weight decay of $0.0005$ and finetune models for 10 epochs on both CIFAR-10 and ImageNet. We set the batch size of 128 and 1024 on CIFAR-10 and ImageNet, respectively. On CIFAR-10, finetune without and with SWAG, we set $\lambda_{\varepsilon,\sigma} = 2$ and $\lambda_{\varepsilon,\sigma} = 0.2$, respectively. On ImageNet, we always set $\lambda_{\varepsilon,\sigma} = 1$. When performing attacks, we set $\sigma = 0.009$ and $\sigma = 0.002$ for models finetuned without and with SWAG, respectively. SWAG rescale the covariance matrix by a constant factor, for disassociate the learning rate from the covariance (Maddox et al., 2019). Here we use $1.5$ as the rescaling factor on ImageNet. Since we found little difference in success rate between using diagonal matrix and diagonal matrix plus low rank matrix as the covariance for SWAG posterior, we always use the diagonal matrix for simplicity. For compared competitors, we followed their official implementations and evaluated on the same test images on CIFAR-10 and ImageNet for generating adversarial examples fairly. Implementation details about these methods are deferred to Appendix D All experiments are performed on an NVIDIA V100 GPU.

## 4.2 COMPARISON TO STATE-OF-THE-ARTS

We compare our method to recent state-of-the-arts in Table 2 and 3 on attacking 10 victim models on ImageNet and 6 victim models on CIFAR-10. Methods that increase input diversity, *i.e.*, and TIM (Dong et al., 2019), SIM (Lin et al., 2019), Admix (Wang et al., 2021), that modify back-propagation, *i.e.*, LinBP (Guo et al., 2020) and ILA++ (Guo et al., 2022), and very recent methods including TAIG (Huang & Kong, 2022) and LGV (Gubri et al., 2022b) are compared. It can be observed in the tables that our method outperforms all these methods significantly. Specifically, our method achieves an average success rate of nearly $91.33\%$ on ImageNet, which outperforms the sec-

---

[1]https://github.com/bearpaw/pytorch-classification

[2]https://pytorch.org/docs/stable/torchvision/models.html

ond best by 19.05%. When attacking Inception v3 and PyramidNet, which are the most challenging victim models on ImageNet and CIFAR-10, respectively, our method outperforms the second best by 27.36% and 7.43% while outperforms the I-FGSM baseline by 57.52% and 36.23%.

**Comparison to (more) ensemble attacks.** From certain perspectives, our method shows a similar spirit to that of ensemble attacks, thus here we compare more such attacks (Liu et al., 2017; Xiong et al., 2022) empirically. Table 4 provides results of all methods under the $\ell_\infty$ constraint and $\epsilon = 8/255$. Our method taking ResNet-50 as the substitute architecture achieves an average success rate of 91.33% on ImageNet, while the two prior methods taking ResNet-50, Inception v3, MobileNet, and MNASNet altogether as substitute architectures achieve 46.58% and 57.52%. One may also utilize more than one architecture for our method, as a natural combination of our method and prior methods, see Table 4 for results. We use ResNet-50 and MobileNet as substitute architectures when combining with the ensemble methods, though more architectures lead to more powerful attacks.

Table 4: Comparison to ensemble attacks on ImageNet under the $\ell_\infty$ constraint with $\epsilon = 8/255$. "Average" is obtained on all the 7 models. We performed 10 runs and report the average results.

| Method | VGG-19 | ResNet-152 | DenseNet | SENet | ResNeXt | WRN | PNASNet | Average |
|---|---|---|---|---|---|---|---|---|
| Ensemble (2017) | 63.68% | 48.14% | 56.42% | 35.96% | 45.36% | 44.06% | 32.46% | 46.58% |
| SVRE (2022) | 73.12% | 60.00% | 66.50% | 46.44% | 57.72% | 55.80% | 43.08% | 57.52% |
| Ours | 97.79% | 97.13% | 98.02% | 85.41% | 94.16% | 95.39% | 77.60% | 92.21% |
| Ours + Ensemble | 98.52% | 97.24% | 98.83% | 87.48% | 94.26% | 95.75% | 79.72% | 93.11% |
| Ours + SVRE | **98.74%** | **97.69%** | **99.04%** | **88.27%** | **95.60%** | **96.24%** | **80.25%** | **93.69%** |

**Comparison to Gubri et al. (2022a)'s method.** Gubri et al. (2022a)'s method suggested to collect multiple substitute models along a single run of training using cSGLD (Zhang et al., 2019). It is related to Bayesian learning in the sense that cSGLD is a Bayesian posterior sampling method. The difference between our method and Gubri et al.'s lie in three main aspect. First, our method is motivated by the belief that scaling the number of substitute models, *to infinity if possible*, improves the transferability of adversarial examples; thus, by establishing a proper posterior with or without finetuning, our method is capable of producing different sets of substitute models at different iterations of I-FGSM, as if there exist infinitely many models (see Appendix C for the benefit in scaling the number of models). By contrast, Gubri et al.'s method utilizes a fixed finite set of models collected during a single run of finetuning. Second, as have been demonstrated in Section 3.2, our method is capable of achieving superior transferability even without finetuning, while finetuning is requisite for Gubri et al.'s method. Third, the derivation in Section 3.2 leads to a principled finetuning objective for our method, which is also strikingly different from that of Gubri et al.'s method. Table 5 compares the two methods empirically and shows the superiority of our method in experiments. We followed their experimental settings to evaluate the methods and copy their results in the paper. More specifically, the back-end I-FGSM was perform with $\epsilon = 4/255$ and a step size of $\epsilon/10$ for 50 optimization steps in total. Victim models include ResNeXt-50, DenseNet-121, EfficientNet-B0 (Tan & Le, 2019), etc.

Table 5: Comparison to Gubri et al. (2022a)'s on ImageNet under the $\ell_\infty$ constraint with $\epsilon = 4/255$. We performed 10 runs and report the average performance for all our results.

| Method | ResNet-50 | ResNeXt-50 | DenseNet | MNASNet | EfficientNet |
|---|---|---|---|---|---|
| Gubri et al. (2022a)'s | 78.71% | 65.11% | 61.49% | 51.81% | 31.11% |
| Ours | **97.14%** | **77.93%** | **80.32%** | **82.15%** | **61.15%** |

**Attack robust models.** It is also of interest to evaluate the transferability of adversarial examples to robust models, and we compare the performance of competitive methods in this setting in Table 6. The victim models here include a robust Inception v3 and a robust EfficientNet-B0 collected from the public timm (Wightman, 2019) repository, together with a robust ResNet-50 and a robust DeiT-S (Touvron et al., 2021a) provided by Bai et al. (2021). All these models were trained using some sorts of advanced adversarial training (Madry et al., 2018; Wong et al., 2020; Xie et al., 2020) on ImageNet. We still used the ResNet-50 source model which was trained just as normal and not robust to adversarial examples. Obviously, in Table 6, our method outperforms the others consistently in attacking these models. In addition to adversarial training, we also tested on robust models obtained via randomized smoothing (Cohen et al., 2019), which is one of the most popular methods for achieving certified robustness. In particular, our method achieves an success rate of **29.25%** in

Table 6: Success rates of attacking vision transformers and adversarially trained models on ImageNet using ResNet-50 as the substitute architecture under the $\ell_\infty$ constraint with $\epsilon = 8/255$. We performed 10 runs and report the average results in the table.

| Method | Vision transformers | | | | Robust models | | | |
|---|---|---|---|---|---|---|---|---|
| | ViT-B | DeiT-B | Swin-B | BEiT | Inception v3 | EfficientNet | ResNet-50 | DeiT-S |
| I-FGSM | 4.70% | 5.92% | 5.18% | 3.64% | 11.94% | 9.48% | 9.26% | 10.68% |
| ILA++ (2022) | 9.48% | 21.34% | 14.88% | 11.76% | 15.54% | 30.90% | 10.08% | 11.08% |
| LGV (2022) | 11.18% | 20.02% | 12.14% | 11.66% | 18.00% | 39.06% | 10.56% | 11.50% |
| Ours | **21.66%** | **43.53%** | **21.84%** | **29.78%** | **25.89%** | **67.05%** | **11.02%** | **12.02%** |

attacking a smoothed ResNet-50 on ImageNet, following the same settings as in Table 6, while I-FGSM, ILA++, and LGV achieve 9.74%, 16.06%, and, 17.64%, respectively.

**Attack vision transformers.** The community has witnessed a surge of transformers in computer vision and machine learning applications. Here we also test the transferability of adversarial examples (generated on convolutional substitute architectures) to some vision transformers. Specifically, we tested with a ViT-B (Dosovitskiy et al., 2020), a DeiT-B (Touvron et al., 2021b), a Swin-B (Liu et al., 2021), and a BEiT(Bao et al., 2021). These models were all collected from timm (Wightman, 2019). Results in Table 6 show that, though it is indeed more challenging to transfer to vision transformers, our method can also gain considerable improvements.

### 4.3 Combination with Other Methods

We would also like to mention that it is also possible to combine our method with some methods in Table 2 to further enhance the transferability, since our method focuses on model diversity and does not consider improvement in input diversity and backpropagation. In Table 7, we report the attack success rate of our method, in combination with Admix, LinBP, and ILA++. It can be seen that the transferability to all victim models are further enhanced. The best performance is obtained when combined with ILA++, leading to an average success rate achieve of 94.65% (which is roughly 32% higher than the original performance of ILA++) and a worst success rate of only 84.44% (when attacking PNASNet).

Table 7: Combining our method with some recent state-of-the-arts in Table 2. The experiment was performed with $\epsilon = 8/255$ on ImageNet. We performed 10 runs and report the average results.

| | Admix | | LinBP | | ILA++ | |
|---|---|---|---|---|---|---|
| | - | + Ours | - | + Ours | - | + Ours |
| ResNet | **100.00%** | **100.00%** | **100.00%** | **100.00%** | 99.96% | **100.00%** |
| VGG-19 | 57.95% | **97.84%** | 72.00% | **98.28%** | 74.94% | **98.60%** |
| ResNet-152 | 45.82% | **97.55%** | 58.62% | **96.49%** | 69.64% | **97.61%** |
| Inception v3 | 23.59% | **77.15%** | 29.98% | **80.65%** | 41.56% | **87.65%** |
| DenseNet | 52.00% | **98.61%** | 63.70% | **98.55%** | 71.28% | **98.95%** |
| MobileNet | 55.36% | **98.13%** | 64.08% | **97.75%** | 71.84% | **98.65%** |
| SENet | 30.28% | **87.49%** | 41.02% | **87.35%** | 53.12% | **89.43%** |
| ResNeXt | 41.94% | **94.98%** | 51.02% | **92.50%** | 65.92% | **95.48%** |
| WRN | 42.78% | **95.90%** | 54.16% | **94.92%** | 65.64% | **96.72%** |
| PNASNet | 21.91% | **78.68%** | 29.72% | **76.94%** | 44.56% | **84.44%** |
| MNASNet | 52.32% | **98.41%** | 62.18% | **97.21%** | 70.40% | **98.95%** |
| Average | 42.40% | **92.47%** | 52.65% | **92.06%** | 62.89% | **94.65%** |

## 5 Conclusion

In this paper, we have considered diversity in substitute models for performing transfer-based attacks. Specifically, we have developed a Bayesian formulation for performing attacks and advocated possible finetuning for improving the Bayesian model. By simply assuming the posterior to be an isotropic Gaussian distribution or, one step further, a more general Gaussian distribution, our attack can be equivalently regarded as generating adversarial examples on a set of infinitely many substitute models while the time complexity of possible finetuning is just as normal. Extensive experiments have been conducted to demonstrate the effectiveness of our method on ImageNet and CIFAR-10. It has been shown our method outperforms recent state-of-the-arts by large margins in attacking more than 10 convolutional DNNs and 4 vision transformers. The transferability to robust models has also been evaluated. Moreover, we have also shown that the proposed method can further be combined with prior methods to achieve even more powerful adversarial transferability.

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

## A  PERFORMANCE OF POSSIBLE FINETUNING ON IMAGENET

Table 8: Comparing transferability of FGSM and I-FGSM adversarial examples generated on a deterministic substitute model and the Bayesian substitute model (with or without additional finetuning) under the $\ell_\infty$ constraint with $\epsilon = 8/255$ on ImageNet. The architecture of the substitute models is ResNet-50, and "Average" was calculated over all ten victim models except for ResNet-50. We performed 10 runs of the experiment and report the average performance in the table.

| | | fine-tune | ResNet-50 | VGG-19 | ResNet-152 | Inception v3 | DenseNet | MobileNet |
|---|---|---|---|---|---|---|---|---|
| FGSM | - | ✗ | 87.68% | 34.40% | 27.06% | 21.46% | 34.38% | 36.88% |
| | Isotropic | ✗ | 87.02% | 49.94% | 40.14% | 33.82% | 49.86% | 51.16% |
| | | ✓ | 100.00% | 66.70% | 55.62% | 31.04% | 63.50% | 64.84% |
| | +SWAG | ✓ | 96.96% | 77.08% | 64.34% | 54.58% | 77.10% | 78.52% |
| I-FGSM | - | ✗ | 100.00% | 39.22% | 29.18% | 15.60% | 35.58% | 37.90% |
| | Isotropic | ✗ | 98.40% | 70.28% | 58.80% | 47.04% | 70.76% | 70.74% |
| | | ✓ | 100.00% | 93.48% | 90.16% | 51.02% | 90.00% | 89.98% |
| | +SWAG | ✓ | 100.00% | 97.74% | 97.12% | 73.24% | 98.06% | 97.50% |

| | | fine-tune | SENet | ResNeXt | WRN | PNASNet | MNASNet | Average |
|---|---|---|---|---|---|---|---|---|
| FGSM | - | ✗ | 17.84% | 24.46% | 24.78% | 15.50% | 34.40% | 27.12% |
| | Isotropic | ✗ | 28.04% | 37.26% | 37.98% | 24.92% | 48.48% | 40.16% |
| | | ✓ | 38.26% | 49.26% | 52.44% | 27.04% | 61.94% | 51.06% |
| | +SWAG | ✓ | 43.00% | 55.64% | 59.34% | 39.36% | 75.48% | 62.44% |
| I-FGSM | - | ✗ | 17.66% | 26.18% | 27.18% | 12.80% | 35.58% | 27.69% |
| | Isotropic | ✗ | 41.36% | 52.06% | 55.06% | 36.18% | 67.50% | 56.98% |
| | | ✓ | 75.88% | 86.48% | 86.92% | 62.86% | 87.56% | 81.43% |
| | +SWAG | ✓ | 85.44% | 94.14% | 95.36% | 77.58% | 97.12% | 91.33% |

## B  SENSITIVITY OF $\lambda_{\varepsilon,\sigma}$

When finetuning is possible, we have $\lambda_{\varepsilon,\sigma}$ as a hyper-parameter. An empirical study was performed to show how the performance of our method varies along with the value of such a hyper-parameter. We tested with $\lambda_{\varepsilon,\sigma} \in \{0, 0.01, 0.05, 0.1, 0.2, 0.5, 1, 1.5, 2\}$ on ImageNet and show the average attack success rates of attacking ten victim models in Figure 3. It can be seen by increasing the value of $\lambda_{\varepsilon,\sigma}$, the adversarial transferability is improved, while it drastically drop when it is too large. Tuning of such a hyper-parameter can be done coarsely in a logarithmic scale on a validation set on different datasets.

## C  BENEFIT OF SCALING THE NUMBER OF SUBSTITUTE MODELS

Our method is developed based on the belief that utilizing more substitute models should improve the transferability of adversarial examples, and the Bayesian formulation is considered since infinitely many models can be sampled from it in principle. In this section, we evaluate the transferability of adversarial examples crafted on different numbers of substitute models, which are sampled before issuing attacks. Figure B illustrates the results, and it can be observed that using more substitute models can indeed improve the transferability of adversarial examples. Average success rates calculated on the same victim models as in Table 2 are reported.

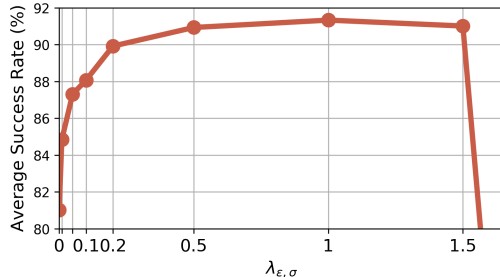 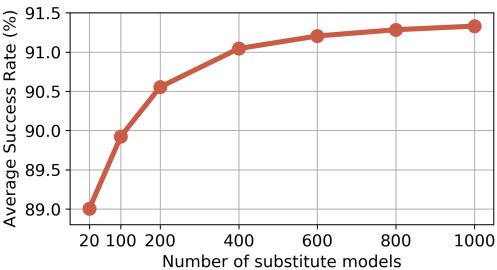

Figure 3: How the transferability changes with the value of $\lambda$ under $\epsilon = 8/255$. Average success rates calculated on the same victim models as in Table 2 are reported.

Figure 4: How the adversarial transferability changes by scaling the number of substitute models under $\epsilon = 8/255$. Average success rates are reported.

## D   DETAILED EXPERIMENTAL SETTINGS OF COMPARED METHODS

Here we report some detailed experimental settings of compared methods. For TIM (Dong et al., 2019), we adopted the maximal shift of $3 \times 3$ and $7 \times 7$ on CIFAR-10 and ImageNet, respectively. For SIM (Lin et al., 2019), the number of scale copies is 5. For Admix (Wang et al., 2021), we set the number of mix copies to 3, and let the mix coefficient be 0.2. For LinBP (Guo et al., 2020), the last six building blocks in ResNet-50 and the last four building blocks in Resnet-18 were modified to be more linear during backpropagation. For TAIG (Huang & Kong, 2022), we set 30 as the number of tuning points. For ILA++ (Li et al., 2020a; Guo et al., 2022), we chose the first block of the last meta-block of ResNet-18 and the first block of the third block of ResNet-50. Ridge regression was adopted for ILA++ since it is faster. Note that as the back-end I-FGSM was perform for 50 optimization steps in total for ImageNet experiments and 20 steps for CIFAR-10 experiments, the results of for instance ILA++ is slightly different from those in its original paper.

