# OpenReview forum: "Making Substitute Models More Bayesian Can Enhance Transferability of Adversarial Examples"
_ICLR.cc/2023/Conference — ICLR 2023 poster_

### Official Review · Reviewer_o774 · 2022-10-21

**Confidence:** 4
**Correctness:** 3
**Technical Novelty And Significance:** 2
**Empirical Novelty And Significance:** 2
**Recommendation:** 5

**Clarity, Quality, Novelty And Reproducibility:**

The idea of using BNNs to craft transferable adversarial attacks is not completely original and novel. The quality and clarity of the paper could be improved (see Strength and Weaknesses Section). Enough details are given in the paper to reproduce the results.

**Strength And Weaknesses:**

Strengths

- The main idea of the paper, which is to use BNNs to obtain transferable adversarial attacks, is clear and intuitive

- Experimental results are convincing and show that the proposed method outperforms competitive algorithms

Weaknesses

- A recent paper [Gubri et al (2022a)] also explored BNNs to obtain transferable attacks. While the methods in this paper seem to obtain better empirical results, the results of this paper feel incremental compared to [Gubri et al (2022a)]

- Section 3 and 4 are often confusing and contain some inaccuracies. In particular:

-- After Eqn 1, p(y | x,w) is the likelihood and not the predictive distribution.

-- In Section 3, experiments are mixed with the explanation of the methods and this creates confusion. Furthermore, in this Section the authors present adversarial attacks for BNNs assuming that the posterior is an isotropic Gaussian. However, for basically all approximate posterior inference methods commonly employed for BNNs, the posterior will in general never be an isotropic Gaussian. In view of this, I would re-organise the Section presenting the method for a general Gaussian posterior.

-- L is Eqn 3 is not defined (I guess it is the loss, but please say it explicitly)

-- In Section 4 the description of the experimental setting takes almost one page. To improve readability I would move some of the details (especially those about the details of the methods the authors compare against) in the appendix.

- Something that is not clear to me is why the authors need to develop a new method to attack BNNs, instead of using methods already available in the literature, see e.g. [1,2,3]. This should be at least discussed and motivated. Also, it is not clear how the authors are attacking BNNs trained with SWAG. Please, clarify

- An important aspect that the authors should keep in mind (and discuss) is that BNNs are known to be more robust to adversarial attacks compared to standard neural networks [4] (depending on the approximate posterior method employed). As a consequence, the results of this paper seems to hint on the fact that while it may be harder to find adversarial attacks for BNNs, these may transfer more easily across different architectures.

- Why in the experiments do you need to use a bigger $\epsilon$ for ImageNet than for Cifar-10? This is counterintuitive for me


[1] Liu, Xuanqing, et al. "Adv-BNN: Improved Adversarial Defense through Robust Bayesian Neural Network." International Conference on Learning Representations. 2018.

[2] Yuan, Matthew, et al. "Gradient-Free Adversarial Attacks for Bayesian Neural Networks." Third Symposium on Advances in Approximate Bayesian Inference. 2020.

[3] Carbone, Ginevra, et al. "Robustness of bayesian neural networks to gradient-based attacks." Advances in Neural Information Processing Systems 33 (2020): 15602-15613.




**Summary Of The Paper:**

The authors consider the problem of crafting adversarial attacks that are transferable across different deep neural networks architectures (DNNs). In order to solve this problem the authors propose to craft the attacks on Bayesian neural networks (BNNs) trained with Gaussian posterior approximations. On a set of experiments on Cifar-10 and ImageNet the authors show that their approach outperforms state-of-the-art methods.

**Summary Of The Review:**

The paper presents an interesting empirical idea with encouraging experimental results. The quality of the paper is hindered by the clarity of the writing, some inaccuracies, and missing discussion and justification about the methods used to train and attack a BNN.

---

> ### Author Response · Authors · 2022-11-17
> **Response to Reviewer o774 (part 1/3)**
>
> Thanks for recognizing the strengths of our paper. Our response to the comments are provided as follows.
>
> >A recent paper [Gubri et al (2022a)] also explored BNNs to obtain transferable attacks.
>
> - As has been discussed in the paper, indeed, both our method and the independent work of Gubri et al (2022a)'s are developed based on a Bayesian spirit. However, the formulation and implementation of both methods are very different.
>
>    - First, our method is motivated by the desire to attacks on **infinitely many** substitute models, given the fact that adversarial examples generated on an ensemble of more source models transfer better, as highlighted in Figure 4, thus our method proposes to learn a Bayesian model and attack it directly for achieving powerful adversarial transferability. By contrast, Gubri et al. (2022a) proposed to keep **a fixed limited number** of model checkpoints during cSGLD training and attack an ensemble of these models.
>
>   - Moreover, the learning objective of our method is formally derived from assumptions of the model prior, while it is also possible to adopt our model even if fine-tuning the source model is impossible (as shown in Table 1 in the paper). By contrast, finetuning is always required for Gubri et al.'s method and its learning objective is also different.
>
>   - Table 5 in our paper shows that these differences lead to significant gains in attacking victim models (at least **+13.55%** on average).
>
> >After Eqn 1, $p(y | \mathbf{x},\mathbf{w})$ is the likelihood and not the predictive distribution.
>
> - Thanks for the comment. It was called the predictive distribution just to keep in line with some prior work [4, 5]. We have revised Section 3 to clearly identify that it is the likelihood.
>
> > In Section 3, experiments are mixed with the explanation of the methods and this creates confusion. Furthermore, for basically all approximate posterior inference methods commonly employed for BNNs, the posterior will in general never be an isotropic Gaussian.
>
> - We agree it can be more reasonable to make more complex assumptions than isotropic Gaussian for commonly known applications of BNNs targeting at, e.g., improved uncertainty estimation [5], larger compression rates [6], etc. However, the aim of our paper is different: to achieve more advanced adversarial transferability. An intuitive principle for achieving better transferability is to encourage the surrogate models to be more similar to target models. While, the distribution of target models is unknown to the adversary (especially without possible finetuning). It should thus be sensible to explore every direction (departed from the mean) of equal importance as the first step, and this motivates us to assume an isotropic Gaussian posterior in Section 3.2 and to defer the exploration of a more general posterior to Section 3.3.
> Experimental results in Section 3.2 are thus provided just to verify the effectiveness of such an isotropic Gaussian assumption empirically. We have added more discussions in the paper to make the motivation and testification of the assumptions clearer.
>
> > $L$ in Eqn 3 is not defined (I guess it is the loss, but please say it explicitly).
>
> - Thanks for pointing it out. Indeed, $L$ in Eq. (3) indicates the loss function, and we have added its definition in the revised paper.
>
> > In Section 4 the description of the experimental setting takes almost one page. To improve readability I would move some of the details (especially those about the details of the methods the authors compare against) in the appendix.
>
> - Thanks for the suggestion. We have moved the introduction to some experiment details to the appendix in the revised paper.

---

> > ### Author Response · Authors · 2022-11-17
> > **Response to Reviewer o774 (part 2/3)**
> >
> > >Something that is not clear to me is why the authors need to develop a new method to attack BNNs, instead of using methods already available in the literature, see e.g. [1,2,3]. This should be at least discussed and motivated. Also, it is not clear how the authors are attacking BNNs trained with SWAG. Please, clarify.
> >
> > - Thanks for pointing out missing related work. We have discussed prior work that studies the robustness of BNNs in the revised paper (particularly in Section 2.2), and we would like to further explain that the aim of our Section 3.1 is to introduce the notations in Bayesian settings and motivate the formulation of our **transfer-based black-box attack** in the Bayesian spirit specifically in Section 3.2 and 3.3, rather than developing a novel **white-box attack** for fooling BNNs as much as possible. In particular, Eq. (2) can be straightforwardly and naturally obtained by adapting previous attacks on deterministic models to Bayesian models, indeed, just like in [1] and [3]. The paper has been revised accordingly to avoid such misunderstandings.
> > For attacking SWAG models, we sampled a fixed number of models from the posterior distribution per attack iteration, and then computed the sign of gradient of the objective function in Eq. (3). Scaled by an update step size of $1/255$, the sign of gradient was adopted to update the model input. As has been mentioned, with iterative optimization, different sets of models can be sampled at different iterations, as if there exist infinitely many substitute models.
> >
> > >BNNs are known to be more robust to adversarial attacks compared to standard neural networks. As a consequence, the results of this paper seems to hint on the fact that while it may be harder to find adversarial attacks for BNNs, these may transfer more easily across different architectures.
> >
> > - Thanks for the suggestion. We have discussed prior work, including [1,2,3,7], that studies the robustness of BNNs in the revised paper, and it has been emphasized in Section 3.2 that more significant transferability can be achieved by introducing the Bayesian formulation to the source model, while the Bayesian model itself is suggested to be more robust. However, we also note that there is no clear evidence of consistent causal relationship between white-box robustness of the substitute model and transferability of the adversarial examples generated on it. In fact, with robust ResNet models obtained via powerful adversarial training [8] on CIFAR-10, one can only achieve 8.18% average success rate (according to our experiments), which is even lower than the baseline result, while our method achieves **83.38%** in the same setting. It has also been demonstrated that, although slight robustness may help achieve improved transferability, further enhancing the robustness of the substitute model can do harm to the transferability of adversarial examples generated on it, (see for instance Figure 8 in [9] in a targeted setting), while the performance of our method improves consistently when sampling more aggressively from the BNNs (see Figure 4 in our paper).
> > The referred paper with a bibliography ID "4" seems missing in your comments. We would be more than delighted to cite it in our paper if it is related.

---

> > > ### Author Response · Authors · 2022-11-19
> > > **Response to Reviewer o774 (part 3/3)**
> > >
> > > > Why in the experiments do you need to use a bigger ϵ for ImageNet than for Cifar-10? This is counterintuitive for me.
> > >
> > > - Experimental results in prior work, e.g., ILA++, show that it is easier to transfer between CIFAR-10 models than between ImageNet models. Given a method, significantly higher attack success rates can be obtained on CIFAR-10 with the same perturbation budget as on ImageNet. Thus, in order to make it as challenging as on ImageNet, we cut the perturbation budget $\epsilon$ by 2x on CIFAR-10 and thus have $\epsilon=4/255$. Table 5 in the paper provides some ImageNet results under $\epsilon=4/255$ as well, following the experimental setting in Gubri et al.'s work. We shall also report the performance of more methods under $\epsilon=4/255$ on more victim models if possible.
> > >
> > > &nbsp;
> > > &nbsp;
> > > &nbsp;
> > >
> > > [1] Liu, Xuanqing, et al. "Adv-bnn: Improved adversarial defense through robust bayesian neural network." International Conference on Learning Representations. 2018.
> > > [2] Yuan, Matthew, Matthew Wicker, and Luca Laurenti. "Gradient-free adversarial attacks for bayesian neural networks." Third Symposium on Advances in Approximate Bayesian Inference. 2020.
> > > [3] Carbone, Ginevra, et al. "Robustness of bayesian neural networks to gradient-based attacks." Advances in Neural Information Processing Systems. 2020.
> > > [4] Izmailov, Pavel, et al. "What are Bayesian neural network posteriors really like?." International conference on machine learning. 2021.
> > > [5] Lakshminarayanan, Balaji, Alexander Pritzel, and Charles Blundell. "Simple and scalable predictive uncertainty estimation using deep ensembles." Advances in neural information processing systems. 2017.
> > > [6] Louizos, Christos, Karen Ullrich, and Max Welling. "Bayesian compression for deep learning." Advances in neural information processing systems. 2017.
> > > [7] Cardelli, Luca, et al. "Statistical Guarantees for the Robustness of Bayesian Neural Networks." International Joint Conference on Artificial Intelligence. 2019.
> > > [8] Zhang, Hongyang, et al. "Theoretically principled trade-off between robustness and accuracy." International conference on machine learning. ICML. 2019.
> > > [9] Springer, Jacob, Melanie Mitchell, and Garrett Kenyon. "A little robustness goes a long way: Leveraging robust features for targeted transfer attacks." Advances in Neural Information Processing Systems. 2021.

---

> > > > ### Author Response · Authors · 2022-11-19
> > > > **Any further questions?**
> > > >
> > > > Thanks again for your comments! Is there any remaining concerns about our paper? We are more than delighted to address any concerns/questions you may have.

---

### Official Review · Reviewer_8LEh · 2022-10-24

**Confidence:** 4
**Correctness:** 3
**Technical Novelty And Significance:** 3
**Empirical Novelty And Significance:** 3
**Recommendation:** 8

**Clarity, Quality, Novelty And Reproducibility:**

Clarity: The writing is clear and the paper is easy to follow.
Quality: The experimental results are impressive.
Novelty: Somewhat novel.
Reproducibility: Not applicable.


**Strength And Weaknesses:**

Strengths:
1.	Utilizing Bayesian models to increase generative adversarial sample mobility is instructive, and the study of ensemble attacks is interesting. The attack method can be equivalently regarded as performing adversarial attack on a set of infinitely many substitute models.
2.	The baselines are advanced, and extensive experiments demonstrate the effectiveness of the proposed method.

Weaknesses:
1.	Some experimental settings are not very reasonable. For example, in Table 4, the authors adopt ResNet-50, Inception v3, MobileNet, and MNASNet altogether as substitute architectures of baselines while adopting ResNet-50 and MobileNet as substitute architectures when combining their method with the ensemble methods. Why not experiment with the same surrogate model? This comparison may be unfair when most of the black box models use the ResNet structure. Moreover, it will be more convincing if the author can add some experiments on advanced defense models, such as adversarial training models, Randomized Smoothing (RS)[1], and Neural Representation Purifier (NRP)[2].
2.	There are some severe spelling mistakes in the article that the author should check carefully, such as “Aidmix” should be “Admix”.
3.	The authors could try to add some analysis on why this method works.
[1]Cohen J, Rosenfeld E, Kolter Z. Certified adversarial robustness via randomized smoothing[C]//International Conference on Machine Learning. PMLR, 2019: 1310-1320.Naseer M, Khan S, Hayat M, et al.
[2] A self-supervised approach for adversarial robustness[C]//Proceedings of the IEEE/CVF Conference on Computer Vision and Pattern Recognition. 2020: 262-271.


**Summary Of The Paper:**

This paper proposes to optimize for diversity in substitute models and advocate attacking a Bayesian model for improving the transferability of adversarial examples.  The author also developed a Bayesian formulation for performing attacks and advocated possible finetuning for improving the Bayesian model.  Extensive experimental results have demonstrated that the proposed method can enhance the transferability of adversarial examples, yielding better attack success rates.

**Summary Of The Review:**

The experimental results are impressive and the idea is interesting.

---

> ### Author Response · Authors · 2022-11-17
> **Response to Reviewer 8LEh**
>
> Thanks for your positive feedback. Our response to the comments are provided as follows.
>
> > Substitute architectures in Table 4.
>
> - We utilized a subset of surrogate architectures only since it takes less time to perform our experiment on fewer surrogate models. We appreciate the suggestion and have tried adopting all four substitute architectures (i.e., ResNet-50, Inception v3, MobileNet, and MNASNet) for performing attacks after the submission deadline, and we observed that even better average success rates could be achieved (90.97%->**91.85%** for our method combined with the ensemble attack and 91.54%->**92.89%** for our method combined with SVRE). Experimental results have been updated in the paper.
>
> > More experiments on advanced defense models.
>
> - As suggested, we have compared our method to state-of-the-arts on defense models. The four robust models in Table 6, i.e., the robust Inception v3, robust EfficientNet, robust ResNet-50, and robust DeiT-S, were all trained via some sort of adversarial training. In addition to the results in Table 6, we further consider randomized smoothing, and the results are given as follows. It can be seen that the superiority of our method holds. These results have been included in Section 4.2. Comparisons of adversarial transferability involving more defense will be provided in future work if possible, considering limited time of the rebuttal period.
> | I-FGSM | ILA++  | LGV    | Ours   |
> |--------|--------|--------|--------|
> | 9.74%  | 16.06% | 17.64% | **29.08%** |
>
> >Some spelling mistakes.
>
> - Thanks for pointing it out, we have revised the paper accordingly to fix them.
>
> >Add some analysis on why this method works.
>
> - More analysis has been included in the paper. Our method is motivated by the belief that generating adversarial examples on an ensemble of more substitute models helps achieve more powerful transferability (than generating on a single substitute model), based on which we conjecture that scaling the number of substitute models to even infinity (if possible) should be more effective. Since the Bayesian formulation learns a distribution of models that are all capable of making reasonable predictions, a Bayesian model can then be utilized to sample all these models as desired. In order to verify the hypothesis, we further performed an experiment to compare the transferability of adversarial examples generated on different numbers of substitute models, all sampled from the learned posterior. The results have been added to the paper in Appendix C, and it can be seen that, indeed, by scaling the number of substitute models, more powerful attack is achieved.

---

> > ### Author Response · Authors · 2022-11-19
> > **Any further questions?**
> >
> > Thanks again for your comments! Is there any remaining concerns about our paper? We are more than delighted to address any concerns/questions you may have.

---

### Official Review · Reviewer_uSCB · 2022-11-02

**Confidence:** 3
**Correctness:** 3
**Technical Novelty And Significance:** 2
**Empirical Novelty And Significance:** 3
**Recommendation:** 6

**Clarity, Quality, Novelty And Reproducibility:**

The Clarity is good. The proposed way to employ Bayesian estimation for attack transferability looks novel. The authors claimed that the code will be available.

**Strength And Weaknesses:**

Strength

1. In general the paper is well-written.
2.  The authors conduct extensive experiments to validate the transferability of the proposed method by concerning many different neural architectures and multiple baselines.
3. The proposed way to employ Bayesian estimation for improving the transferability of attacks looks novel to me.

Weaknesses

1. When both the substitute model and the victim model are adversarially trained (Table 6), the improvement is quite marginal. Therefore, the proposed method may only be employed to improve the transferability of attacks on nonrobust models, which weakens the empirical significance.
2.  In Eq. 6, ∆w is sampled from the gaussian prior. Then why do we need p(∆w) ≥ ε? Plus, if p(∆w) ≥ ε is important, then ε should be a very important hyper-parameter but there is no experiment showing the sensitivity,


**Summary Of The Paper:**

This paper proposes to attack a Bayesian model for improving the transferability of black-box adversarial attacks. Specifically, the authors employ gradient-based attack algorithms on their constructed Bayesian models and expect the generated adversarial attacks can better fool other unseen models. Extensive experiments show that the proposed method surpassed baseline methods in terms of attack success rate.

**Summary Of The Review:**

Given the strength and weaknesses, I tend to rate the paper as marginally above the acceptance.

---

> ### Author Response · Authors · 2022-11-17
> **Response to Reviewer uSCB**
>
> Thanks for your positive feedback. Our response to the comments are provided as follows.
>
> >When both the substitute model and the victim model are adversarially trained (Table 6), the improvement is quite marginal. Therefore, the proposed method may only be employed to improve the transferability of attacks on nonrobust models, which weakens the empirical significance.
>
> - We would like to point out politely that the substitute model for obtaining Table 6 is actually a normally trained ResNet-50. Thus, it is challenging from its nature to transfer such adversarial examples to robust models, as robust and non-robust models can be very different. In addition, we adopted a small perturbation radius with $\epsilon=8/255$, which is different from some prior work that applied $\epsilon=16/255$ [1,2,3], making the transferability even more challenging. While, under such a challenging circumstance, our method still achieves +3.50% absolute gain in attacking the robust Inception v3, +16.86% in attacking the robust EfficientNet, and **+5.34%** absolute gain on average (when compared to the second best), which should be considerable gains in such a challenging setting to the best of our knowledge.
>
> - When attacking with $\epsilon=16/255$, we achieve the below results (**+12.34%** absolute gain on average when compared to the second best), and even more significant gains should be achieved if a robust DNN is utilized as the substitute model.
> |       | Inception v3 | EfficientNet | ResNet-50 | DeiT-S |
> |:-------:|:--------------:|:--------------:|:-----------:|:--------:|
> | IFGSM | 18.62%       | 23.68%       | 12.74%    | 12.42% |
> | ILA++ | 26.86%       | 57.72%       | 13.82%    | 13.74% |
> | LGV   | 27.42%       | 69.84%       | 15.58%    | 14.54% |
> | Ours  | 44.78%       | 95.70%       | 19.40%    | 16.86% |
>
> - In addition to robust models obtained via adversarial training, we also report the performance of attacking a smoothed ResNet which shows certified robustness [4]. It shows that our method achieves a success rate of **29.08%**, while I-FGSM, ILA++, and LGV achieve 9.74%, 16.06%, and 17.64%, respectively. The results further verify the effectiveness of our method in attacking robust models.
>
> > In Eq. 6, $\Delta \mathbf{w}$ is sampled from the gaussian prior. Then why do we need $p(\Delta \mathbf{w}) \geq \varepsilon$?
>
> - In Eq. (6), the maximal empirical loss is penalized in the inner optimization of Eq. (6). Without the constraint of $p(\Delta \mathbf{w}) \geq \varepsilon$, infinitely large loss could be obtained (considering that the Gaussian random variable is unbounded), thus the constraint is necessary and this indicates sampling from a region of high confidence. Note that, with Taylor’s theorem for approximation, optimizing the maximal loss in Eq. (6) is functionally equivalent to optimizing the expected loss.
>
> > If $p(\Delta \mathbf{w}) \geq \varepsilon$ is important, then $\varepsilon$ should be a very important hyper-parameter but there is no experiment showing the sensitivity.
>
> - After applying Taylor’s theorem, we obtain Eq. (7) which has an analytic solution, and then $\sigma$ and $\epsilon$ are reparameterized into a single hyper-parameter $\lambda_{\epsilon, \sigma}$. We have tested the sensitivity of our method to different $\lambda_{\epsilon, \sigma}$ values and the results are given in Appendix B. We appreciate the suggestion!
>
> &nbsp;
> &nbsp;
> &nbsp;
>
> [1] Dong Y, Pang T, Su H, et al. Evading defenses to transferable adversarial examples by translation-invariant attacks. CVPR 2019.
> [2] Lin J, Song C, He K, et al. Nesterov Accelerated Gradient and Scale Invariance for Adversarial Attacks. ICLR 2019.
> [3] Wang X, He X, Wang J, et al. Admix: Enhancing the transferability of adversarial attacks. CVPR 2021.
> [4] Cohen J, Rosenfeld E, Kolter Z. Certified adversarial robustness via randomized smoothing. ICML 2019.

---

> > ### Author Response · Authors · 2022-11-19
> > **Any further questions?**
> >
> > Thanks again for your comments! Is there any remaining concerns about our paper? We are more than delighted to address any concerns/questions you may have.

---

### Decision · Program_Chairs · 2023-01-20

**Decision:**

Accept: poster

**Justification For Why Not Higher Score:**

This is a good paper overall, with solid technical contributions and extensive experiments. However, AC thinks that the paper doesn't bring new insights into adv. attack and adv. sample generation.

**Justification For Why Not Lower Score:**

The contributions of the paper are above the bar of ICLR, and the paper should be of broad interest of ICLR audience.

**Metareview: Summary, Strengths And Weaknesses:**

This paper proposes to attack a Bayesian model for improving the transferability of black-box adversarial attacks. A gradient-based attack is constructed where the generated adversarial attacks can better fool other unseen models. Extensive experimental results have demonstrated that the proposed method can enhance the transferability of adversarial examples, yielding better attack success rates.

Strength:
1. Well-written presentation
2. A novel Bayesian attack method
3. Extensive experiments on adversarial transferability

Weakness:
1. Need more discussions on most recent works
2. Some of the experimental settings are unclear

**Note From Pc:**

if the above contains the word "oral" or "spotlight" please see: "oral" presentation means -> notable-top-5% and "spotlight" means -> notable-top-25%. As stated in our emails, we are disassociating presentation type from AC recommendations